# Metagenomic Analysis of Raw Milk and the Inactivation of Foodborne Pathogens Using Ultraviolet-C

**DOI:** 10.3390/foods14081414

**Published:** 2025-04-19

**Authors:** Ju-Hui Lee, Hyeonjun Moon, Hye-Rim Park, Ji-In Noh, Sang-Soon Kim

**Affiliations:** 1Department of Food Engineering, Dankook University, Cheonan 31116, Republic of Korea; yamaha2011@naver.com (J.-H.L.); moonhj1102@naver.com (H.M.); mayrim@dankook.ac.kr (H.-R.P.); 9627hdc@naver.com (J.-I.N.); 2School of Animal & Food Sciences and Marketing, Konkuk University, Seoul 05029, Republic of Korea

**Keywords:** raw milk, metagenomic analysis, UV-C, *Pseudomonas* spp., resistance

## Abstract

The purpose of this study was to identify the microbial community of raw milk samples before and after UV-C irradiation and to establish fundamental data on UV-C treatment to improve the safety and shelf life of raw milk. Metagenomic analysis revealed that *Lactococcus* spp., *Lactobacillus* spp., and *Staphylococcus* spp. were the dominant genera in raw milk, while *Pseudomonas* spp. became more prevalent after 14 days of refrigerated storage. The microorganisms in raw milk were isolated using selective media and identified as *Serratia quinivorans* 4364 and *Latilactobacillus curvatus* DSM 20019. To compare the UV resistance of these microorganisms, *Pseudomonas aeruginosa*, *Staphylococcus aureus*, *Lactococcus lactis*, and *Latilactobacillus curvatus* were inoculated into sterilized milk and subjected to UV-C treatment. The reduction rates of *P*. *aeruginosa* were significantly lower than those of the other strains (*S*. *aureus*, *L*. *lactis*, and *L*. *curvatus*). These findings provide insights into the microbial distribution in raw milk and the degree of resistance to UV treatment, which can serve as fundamental data for the pasteurization of raw milk.

## 1. Introduction

The quality of pasteurized milk is influenced by the quality of raw milk, processing conditions, and storage and distribution temperatures [1]. Therefore, improving the quality of raw milk is essential for enhancing the quality of pasteurized milk provided to consumers [2]. The microbiological quality of milk can be assessed by the level of psychrotrophic bacteria in raw milk [3]. Recently, the installation of cooling systems on farms and the introduction of a cold chain system have improved the microbiological quality of raw milk. However, the storage of raw milk below 4 °C prior to heat processing might lead to the proliferation of psychrotrophic bacteria, which can grow at temperatures between 2 and 7 °C, negatively affecting milk quality [4]. The most commonly detected psychrotrophic bacteria in raw milk are Gram-negative bacteria, such as *Pseudomonas* spp., *Acinetobacter* spp., *Enterobacter* spp., *Flavobacterium* spp., and *Serratia* spp. [5,6]. These bacteria can be eliminated through pasteurization or ultra-high temperature (UHT) treatment; however, they produce heat-resistant enzymes, such as proteases and lipases, during refrigerated storage [7]. These enzymes are the primary spoilage enzymes affecting the quality of dairy products. Lipases hydrolyze milk triglycerides, leading to lipid oxidation and rancidity, which results in flavor deterioration in products such as cheese and milk. Proteases hydrolyze casein fractions, contributing to milk gelation and bitterness [8,9]. Therefore, controlling the initial load of psychrotrophic bacteria is essential to minimize the negative impact of these spoilage enzymes on dairy product quality [10].

Ultraviolet (UV) radiation is known to be an effective non-thermal pasteurization technology for controlling pathogenic bacteria and fungi in various foods. It is considered a cost-effective and easily accessible method compared to other non-thermal treatment technologies [11]. UV radiation spans a wavelength range of 100–400 nm, falling into three wavelength zones: UV-A (315–400 nm), UV-B (280–315 nm), UV-C (200–280 nm), and vacuum-UV (100–200 nm). The UV-C region is known to provide the most effective inactivation against a wide range of microorganisms [12]. UV-C induces DNA damage by forming pyrimidine dimers within the DNA of microorganisms, which prevents microbial replication and, ultimately, leads to cell death [13]. Previous studies have reported that UV-C light can effectively reduce pathogenic bacteria in raw milk and dairy products. B. Engin and Y. Karagul Yuceer (2012) demonstrated that UV-C treatment resulted in a significant reduction of approximately 2 log CFU/mL in the total coliform count, in *Escherichia coli*, and in *Staphylococcus* spp. in raw milk, making it as effective as a thermal treatment [14]. Similarly, the study by Crook, J. et al., (2012) evaluated the effect of UV-C treatment on the inactivation of seven milk-borne pathogens. Among these, *Listeria monocytogenes* exhibited the highest resistance to UV-C, while *Staphylococcus aureus* was the most susceptible [15]. To the best of our knowledge, no studies have yet investigated the effects of UV-C treatment on the reduction of *Pseudomonas* and lactic acid bacteria in milk.

Next-generation sequencing (NGS), a DNA-based technology, offers a high-throughput and rapid approach for detecting and identifying microorganisms associated with samples. NGS-based metagenomic analysis has been applied in various research fields, including the food industry [16]. In food production, it is used to investigate changes in microbial communities during raw material processing and manufacturing, helping to evaluate microbial diversity and monitor foodborne pathogens [17]. Additionally, in fermented foods, metagenomic analysis has been utilized to identify key microbial communities during fermentation, enabling the optimization of fermentation conditions and the enhancement of product quality [18]. The general workflow of metagenomic analysis includes sample collection, DNA extraction, library preparation, and data analysis [19]. Among various metagenomic analyses, amplicon sequencing of the 16S rRNA gene is a cost-effective method to elucidate the diversity and complexity of bacterial communities [20]. The bacterial 16S rRNA gene contains nine variable regions (V1–V9) interspersed between conserved sequences across different taxa [21]. For bacterial identification, the 16S rRNA gene is first amplified using polymerase chain reaction (PCR) with primers annealing to the conserved regions, followed by sequencing [22]. Sequencing data undergo bioinformatic analysis, with variable regions being utilized to distinguish bacterial taxa [23].

Therefore, in the present study, 16S rRNA amplicon sequencing was conducted to analyze the metagenome of raw milk, which was stored at 4 °C for 14 days of refrigeration and subjected to 5 min of UV-C treatment afterward. Additionally, microorganisms were isolated and identified from raw milk, and the dominant species identified through metagenomic analysis, along with the isolated strains, were used as experimental strains and inoculated into sterilized milk. The resistance of the isolated strains was evaluated using a UV irradiation device.

## 2. Materials and Methods

### 2.1. Sample Preparation

The raw milk samples used in this study were obtained in April 2024 from a dairy farm located in Gwangcheon-eup, Hongseong-gun, Chungcheongnam-do, Republic of Korea. Milking was conducted under identical conditions on six cows, and the samples were randomly divided into three groups, with two cows per group. Milk samples from cows 1 and 2 were labeled as Group A, from cows 3 and 4 as Group B, and from cows 5 and 6 as Group C. All samples were aliquoted into 50 mL and stored at 4 °C. A total of three types of samples were used for metagenomic analysis: raw milk immediately after milking, raw milk refrigerated for 14 days, and raw milk refrigerated for 14 days, followed by 5 min of UV-C treatment. Sterilized milk samples were purchased from Dankook University in Cheonan-si, Chungcheongnam-do, Republic of Korea, and were stored at 4 °C until they were used in the experiment.

### 2.2. DNA Extraction and 16S rRNA Gene Library Construction

Genomic DNA was extracted using the DNeasy^®^ PowerFood^®^ Microbial Kit (Qiagen, Hilden, Germany) according to the manufacturer’s instructions. Three samples were extracted in triplicate, yielding a total of nine DNA samples. The extracted DNA was quantified and quality-controlled by means of 1% agarose gel electrophoresis. Following DNA quality verification, the V3–V4 variable region of the 16S rRNA gene was amplified for sequencing using forward and reverse primers. The sequences of primers were as follows:

(341F:5′-TCGTCGGCAGCGTCAGATGTGTATAAGAGACAGCCTACGGGNGGCWGCAG-3′)

(806R:5′-GTCTCGTGGGCTCGGAGATGTGTATAAGAGACAGGACTACHVGGGTATCTAATCC -3′)

Amplified free primers and primer dimer species were purified using AMPure XP beads (Cat. No. A63881; Beckman Coulter, Pasadena, CA, USA). Subsequently, dual indices and Illumina sequencing adapters were attached using the Nextera XT Index Kit (Illumina, San Diego, CA, USA), and the amplicons were further purified with AMPure XP beads. Prior to sequencing, the DNA concentration of each PCR product was determined using a Qubit 3.0 Fluorometer, and quality control was performed using a Bioanalyzer (Agilent 2100, Santa Clara, CA, USA). A total of nine genome libraries were prepared for single Illumina sequencing. Sequencing was conducted using the Illumina NextSeq system (Illumina, San Diego, CA, USA) according to the manufacturer’s instructions.

### 2.3. Sequencing Data Processing and Bioinformatics Analysis

Sequencing data generated by the Illumina NextSeq system (2 × 300 bp) were analyzed using the QIIME2 software package version 2024.02 [24]. For bioinformatic analysis, the Deblur pipeline within QIIME2 was employed to remove chimera sequences and noise, resulting in high-quality sequences. The denoising process involved removing artificial sequences, trimming low-quality reads, and merging high-confidence forward and reverse reads. Initial quality filtering was performed based on quality scores, and sequences were truncated to a length of 280 bp (–p-trim-length 280). Deblur generated amplicon sequence variants (ASVs), which were used for downstream analyses. To compare the richness of operational taxonomic units (OTUs) across samples, the sequencing depth was normalized to 37,217 reads per sample. Taxonomic classification was performed using this standardized dataset. Reference sequences for taxonomic analysis were obtained from the SILVA 138 database, and the classification results were visualized using the Tidyverse package version 2.0.0 in R Studio version 2023.06.0 build 421 [25].

### 2.4. Isolation and Identification of the Microorganisms in Raw Milk

Selective media, including Pseudomonas agar (KisanBio Co., Ltd., Seoul, Republic of Korea), M17 agar (KisanBio Co., Ltd., Seoul, Republic of Korea), and MRS agar (KisanBio Co., Ltd., Seoul, Republic of Korea), were used to isolate and culture the microorganisms from raw milk. Pseudomonas agar was incubated at 25 °C for 48 h, M17 agar at 37 °C for 24 h, and MRS agar at 37 °C for 48 h. A single colony was randomly selected and subcultured at least three times to obtain a pure isolate. For the identification of the isolated colonies, the 16S rRNA experimental methods provided by BioFACT were employed. Bacterial DNA extraction, PCR amplification, purification, and sequencing were conducted according to BioFACT protocols. Universal primers (27F:5′-AGAGTTTGATCCTGGCTCAG-3′) and (1492R:5′-TAC GGYTACCTTGTTACGACTT-3′) were used for PCR amplification, and the purified PCR products were processed using a PCR purification kit (BioFACT Co., Ltd., Daejeon, Republic of Korea). The purified DNA was sequenced using a DNA analyzer (ABI PRISM 3730XL, Applied Biosystems, Foster City, CA, USA). To enhance sequencing accuracy, the BigDye^®^ Terminator v3.1 Cycle Sequencing Kit (Applied Biosystems, Foster City, CA, USA) was used. The obtained DNA sequences were identified using the BLAST tool of the National Center for Biotechnology Information (NCBI).

### 2.5. Bacterial Cell Preparation

The bacterial strains used in this study included *Staphylococcus aureus* KCTC 25923, *Pseudomonas aeruginosa* KCTC 1637, and *Lactococcus lactis* subsp. lactis obtained from the Department of Food Science and Technology at Dankook University (Cheonan, Republic of Korea), as well as the strain *Latilactobacillus curvatus* DSM 20019 isolated from raw milk. *S*. *aureus* and *P*. *aeruginosa* were cultured in 5 mL of tryptic soy broth (TSB; BD Difco, Sparks, MD, USA) at 37 °C for 24 h. *L*. *lactis* and *L*. *curvatus* were cultured in 5 mL of MRS Broth (KisanBio Co., Ltd., Seoul, Republic of Korea) at 37 °C for 48 h and then centrifuged at 4470× *g* for 20 min at 4 °C (Union 55R, Hanil Science Co., Seoul, Republic of Korea). The resulting pellets were resuspended in 9 mL of 0.2% peptone water (PW; BD Difco, Sparks, MD, USA), corresponding to a population level of 7 log CFU/mL.

### 2.6. UV-C Treatment

In sterilized Erlenmeyer flasks, 95 mL of sterilized milk samples were prepared, and 5 mL of each bacterial culture was inoculated to be 6–7 log CFU/mL. The inoculated samples were distributed into Petri dishes (60 mm diameter × 15 mm height, 10060, SPL Life Sciences Co., Ltd., Pocheon, Republic of Korea) in 10 mL portions. For non-thermal sterilization, a UV sterilization lamp (253.7 nm, 20 W, GL20T5, Sankyo Denki Co., Kanagawa, Japan) was used. The distance between the UV-C lamp and the inoculated milk samples was 17 cm, with a UV-C intensity of 1.79 mW/cm^2^. UV-C treatment was performed while stirring the inoculated samples in Petri dishes at 300 rpm using a magnetic stir bar (Stirrer MSH-20D, DAIHAN Scientific Co., Wonju, Republic of Korea). Resistance to UV-C was assessed from 0 to 15 min (3 min intervals) for *P*. *aeruginosa* and from 0 to 14 min (2 min intervals) for *S*. *aureus*, *L*. *curvatus*, and *L*. *lactis*.

### 2.7. Microbial Colony Count via the Standard Plate Method

The microbial cell colony was measured using the standard plate count method. Samples before and after UV-C treatment were serially diluted in 0.2% peptone water and cultured on selective media to determine the viable cell count. Colonies of *P*. *aeruginosa* were counted after culturing on Pseudomonas agar at 25 °C for 48 h. Colonies of *S*. *aureus* were counted after culturing on Baird–Parker agar (KisanBio Co., Ltd., Seoul, Republic of Korea) at 37 °C for 24 h. *L*. *lactis* colonies were counted after culturing on brain heart infusion agar (KisanBio Co., Ltd., Seoul, Republic of Korea) at 30 °C for 48 h. *L*. *curvatus* colonies were counted after culturing on MRS agar at 37 °C for 48 h.

### 2.8. Statistical Analysis

All experiments were conducted in triplicate, and the experimental results were analyzed using a one-way ANOVA in IBM SPSS Statistics 27 (IBM Corp., Armonk, NY, USA). Post hoc analysis was performed using Duncan’s multiple range test to determine significant differences (*p* < 0.05)

## 3. Results and Discussion

### 3.1. Metagenomic Analysis Following UV-C Treatment

One of the limitations of this study was the sample size. Due to practical constraints in the laboratory, obtaining a larger quantity of milk from more cows was not feasible in this study. The limitation was minimized by employing random sampling within the available resources. Specifically, two cows were selected from each group, and for each cow, 200 mL of milk was collected and randomly mixed. This approach was intended to introduce a level of randomness and improve representativeness within the scope of our available resources. The results of the taxonomic analysis of nine raw milk samples using 16S rRNA amplicon sequencing are presented in Figure 1. The nine samples (three samples in triplicate) were categorized into three groups for metagenomic analysis: raw milk immediately after milking (bovine milk-A: BA, -B: BB, and -C: BC), refrigerated raw milk without UV-C treatment (sample-A: SA, -B: SB, and -C: SC), and refrigerated raw milk treated with UV-C (sample-UVA: CA, -UVB: CB, and -UVC: CC). To compare and analyze the microbial distribution in the raw milk immediately after milking, the refrigerated milk, and the UV-C-treated milk, the SILVA database was utilized. At the genus level, the microbial community analysis showed that in BA, *Lactococcus* spp. and *Lactobacillus* spp. were dominant, comprising 40.76% and 39.38%, respectively, followed by *Bifidobacterium* spp. at 17.51%. In BB, *Staphylococcus* spp. was dominant at 64.41%, while BC showed a similar distribution to BA, with *Lactococcus* spp. and *Lactobacillus* spp. comprising 36.31% and 35.96%, respectively.

In refrigerated raw milk stored for 14 days, *Pseudomonas* spp. was dominant, comprising 58.71% in SA, 23.08% in SB, and 29.11% in SC. Compared to fresh raw milk, the refrigerated samples showed a higher proportion of *Pseudomonas* spp., which is consistent with previous studies reporting an increase in *Pseudomonas* spp. during refrigerated storage at 4 °C for up to 7 days [26]. In CB, a refrigerated raw milk sample treated with UV-C, *Pseudomonas* spp. showed an increased proportion of 29.89%, whereas *Lactococcus* spp. decreased to 7.72% compared to the untreated SB. This indicates that *Pseudomonas* spp. exhibit greater resistance to UV-C treatment than *Lactococcus* spp. Therefore, subsequent experiments were conducted to evaluate the UV-C resistance of *Pseudomonas* spp., *Staphylococcus* spp., and *Lactococcus* spp.

### 3.2. Isolation and Identification of Microorganisms in Raw Milk

The 16S rRNA gene sequences of strains isolated from raw milk were analyzed and compared with sequences registered in the NCBI GenBank database. The strain isolated and identified on Pseudomonas agar exhibited high sequence similarity with *Serratia quinivorans* 4364 (98.71%) and *Serratia liquefaciens* ATCC 27592 (98.64%), confirming the strain as *Serratia quinivorans* 4364. The strain isolated and identified on M17 agar showed a high sequence similarity with *Serratia grimesii* DSM 30063 (98.71%) and *Serratia quinivorans* 4364 (98.64%), identifying the strain as *Serratia grimesii* DSM 30063. The strain isolated and identified on MRS agar demonstrated high sequence similarity with *Latilactobacillus curvatus* DSM 20019 (99.8%) and *Latilactobacillus curvatus* NBRC 15884 (99.7%), confirming the strain as *Latilactobacillus curvatus* DSM 20019. Therefore, the strain *Latilactobacillus curvatus* DSM 20019, which was isolated and identified from MRS agar (a selective medium for lactic acid bacteria like *Lactobacillus* spp.), was used in the study as a lactic acid bacterium.

### 3.3. UV-C Resistance

An analysis of the effect of the UV-C treatment on sterilized milk inoculated with *P*. *aeruginosa*, *S*. *aureus*, *L*. *lactis*, and *L*. *curvatus* revealed that *P*. *aeruginosa* exhibited the highest resistance to the UV-C treatment (Table 1 and Figure 2). Bacterial populations decreased over time due to the UV-C treatment, but the extent of reduction varied among the bacterial species. As shown in Table 1, the population of *P. aeruginosa* decreased by approximately 2 log CFU/mL after 6 min of treatment compared to the untreated control, with no significant reduction observed beyond this point. For *S. aureus* and *L. curvatus*, a gradual decrease in population was observed over time; however, no significant reduction was noted after 8 min of treatment. Meanwhile, *L. lactis* exhibited a notable reduction, with its count falling below the detection limit (1 log CFU/mL) after 10 min of treatment.

In summary, UV-C treatment for approximately 3 min resulted in a reduction of about 2 log CFU/mL in *P*. *aeruginosa*, *S*. *aureus*, and *L*. *lactis*, while *L*. *curvatus* showed a reduction of about 1 log CFU/mL. *P*. *aeruginosa* demonstrated a linear decrease curve with UV-C treatment, but after 6 min of treatment, a tailing curve was observed, which is consistent with a previous metagenomic analysis that showed *Pseudomonas* spp. to be more resistant than *Lactococcus* spp. According to this previous study, *P*. *aeruginosa* exhibited a linear decrease in the initial stages of UV-C treatment, followed by a tailing curve after a certain treatment time [27]. However, due to the higher UV-C resistance of *P*. *aeruginosa* compared with lactic acid bacteria, UV-C treatment alone is unlikely to sufficiently ensure the quality of raw milk. The major challenge of UV-C treatment is its limited penetration depth, particularly in opaque or turbid liquids such as defatted milk. The high scattering and absorption coefficients of these fluids make it difficult to achieve a uniform fluence distribution, which can affect microbial inactivation efficiency. Consequently, when implementing UV-C treatment on an industrial scale, careful system design is required to ensure effective exposure. This includes incorporating continuous agitation mechanisms and optimized reactor configurations, such as thin-film flow systems or pulsed UV technology. This study was focused on identifying the major microbial communities through metagenomic analysis and evaluating the effectiveness of UV-C treatment in reducing pathogens. However, other aspects of milk quality, such as pH, protein structure, and/or lipid composition, would be important for pasteurization. Therefore, a comprehensive assessment of milk quality after UV-C treatment would be more beneficial in a further study.

## 4. Conclusions

This study investigated the microbial composition in raw milk using metagenomic analysis. The proportion of *Pseudomonas* spp. increased with extended refrigeration. After 14 days of refrigerated storage, the proportion of *Lactococcus* spp. and *Lactobacillus* spp. decreased in UV-C-treated raw milk compared with the untreated samples. Based on these findings, a non-thermal UV-C treatment experiment was performed using *P*. *aeruginosa*, *L*. *lactis*, and *L*. *curvatus* (isolated from raw milk) to evaluate resistance. The results showed reductions: *S*. *aureus*, *L*. *lactis*, and *L*. *curvatus* exhibited a 1 log CFU/mL reduction after 2 min of UV-C exposure. The population of *P*. *aeruginosa* showed a decreasing trend after 3 min of treatment, but the reduction level was significantly lower than that of other strains (*S*. *aureus*, *L*. *lactis*, and *L*. *curvatus*). These findings suggest that UV-C-based methods hold potential in industrial applications for improving raw milk quality. However, since UV-C treatment alone is insufficient to effectively inhibit *P*. *aeruginosa*, a major spoilage bacterium in raw milk, future research should focus on combination treatments, such as integrating thermal pasteurization, to reduce the prevalence of *Pseudomonas* spp. in raw milk.

## Figures and Tables

**Figure 1 foods-14-01414-f001:**
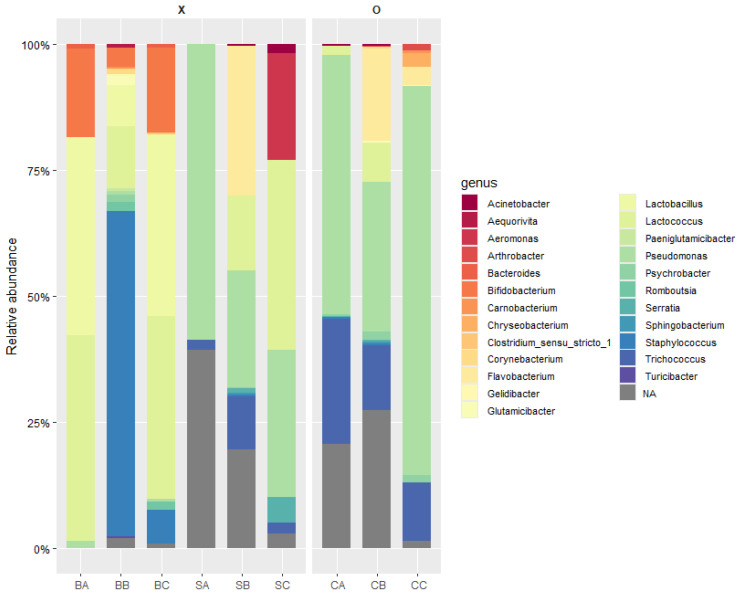
Taxonomic analysis of nine raw milk samples using 16S rRNA amplicon sequencing. The analyzed raw milk samples were grouped as “raw milk (BA, BB, BC)”, “refrigerated raw milk (SA, SB, SC)”, and “UV-C treated raw milk (CA, CB, CC)”.

**Figure 2 foods-14-01414-f002:**
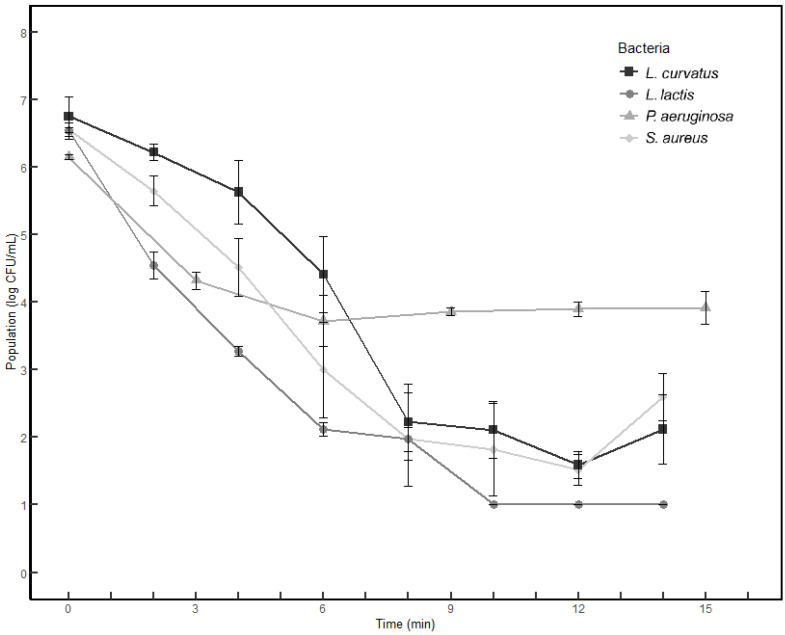
Population of Latilactobacillus curvatus, Lactococcus lactis, Pseudomonas aeruginosa, and Staphylococcus aureus subjected to UV-C irradiation.

**Table 1 foods-14-01414-t001:** UV-C reduction patterns of bacteria inoculated in sterilized milk.

*P. aeruginosa*		*S. aureus*		*L. lactis*		*L. curvatus*	
Time (min)	Log_10_(N)	Time (min)	Log_10_(N)	Time (min)	Log_10_(N)	Time (min)	Log_10_(N)
0	6.15 ± 0.03 ^a^	0	6.55 ± 0.03 ^a^	0	6.54 ± 0.12 ^a^	0	6.75 ± 0.29 ^a^
3	4.32 ± 0.13 ^b^	2	5.65 ± 0.22 ^a^	2	4.54 ± 0.20 ^b^	2	6.22 ± 0.12 ^ab^
6	3.72 ± 0.38 ^c^	4	4.52 ± 0.43 ^b^	4	3.28 ± 0.07 ^c^	4	5.63 ± 0.47 ^b^
9	3.86 ± 0.06 ^bc^	6	3.00 ± 0.71 ^c^	6	2.12 ± 0.10 ^d^	6	4.41 ± 0.51 ^c^
12	3.90 ± 0.11 ^bc^	8	1.97 ± 0.69 ^de^	8	1.97 ± 0.18 ^d^	8	2.23 ± 0.56 ^d^
15	3.92 ± 0.24 ^bc^	10	1.82 ± 0.69 ^de^	10	ND	10	2.11 ± 0.42 ^d^
		12	1.52 ± 0.23 ^e^	12	ND	12	1.59 ± 0.20 ^d^
		14	2.60 ± 0.35 ^cd^	14	ND	14	2.12 ± 0.52 ^d^

Different letters represent significant differences between time points within the same bacterial strain (*p* < 0.05). ND: not detected.

## Data Availability

The original contributions presented in this study are included in the article. Further inquiries can be directed to the corresponding author.

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
