# Peer review of "Metagenomic Analysis of Raw Milk and the Inactivation of Foodborne Pathogens Using Ultraviolet-C"

_foods, 2025, doi:10.3390/foods14081414_

Round 1
Reviewer 1 Report
Comments and Suggestions for Authors
In abstract, the aim, method, results and key findings are not well presented.
- In introduction
A weak discussion on how psychrotrophic bacteria can cause either quality or safety issues to raw milk. Missed some key references, International Journal of Dairy Technology, 76(1):51-62, DOI: 10.1111/1471-0307.12905; Journal of Dairy Science 107:1950–1966, DOI: 10.3168/jds.2023-23860; Journal of Food Protection, 2019, DOI: 10.4315/0362-028X.JFP-19-032.
For Ultraviolet (UV) radiation, give any references to support its inactivation in pathogenic bacteria and fungi;
Next-generation sequencing (NGS), too much description of its basis information. I would suggest the authors to provide exact applications in the food industry.
- Materials and Methods
In this part , the authors missed some key ref to support.
“raw milk refrigerated for 14 days”, in my opinion, the refrigeration time is too long, 14 d is not widely used in the industry.
2.4. Isolation and identification of microorganisms in raw milk, the culture conditions are not clear, like incubation time, temperature.
UV-C Treatment, any referece to support the parameters?
- Results
Remove any reference in this part to the discussion, simply clarify the data.
Carefully check the writing of each bacterial name.
Only Latilactobacillus curvatus DSM 20019 was sequenced in this study, need to be clarified.
Though the authors stated that Post- 172 hoc analysis was performed using Duncan's multiple range test to determine significant differences (p < 0.05), but actually they missed the significant differences analysis in this part.
Discussion part is missed in this manuscript.
Comments on the Quality of English LanguageNone
Author Response
We are sincerely grateful for your thorough consideration and scrutiny of our manu-script, “Metagenomic analysis of raw milk and the inactivation of foodborne pathogen using ultraviolet-C”, control number foods-3547901. The manuscript was edited by MDPI author services and certificate was attached.
We have carefully considered all the feedback and made the necessary revisions to the manuscript. Our point-by-point responses to each of the reviewers’ comments are provided below. The reviewers' comments are highlighted in blue, our responses are in black, and the changes to the manuscript are highlighted in red for your convenience. We have made every effort to address all of the reviewers’ concerns and hope that the revised manuscript is now acceptable for publication in Foods.
Reviewer #1
1) Comment: In abstract, the aim, method, results and key findings are not well presented.
Response: We appreciate the reviewer’s comment. We have carefully revised the abstract to clarify the aim of the study. Given the concise nature of the abstract, we have maintained a succinct description of the methods and results while ensuring that the purpose of the study is clearly articulated. We have addressed this point in the abstract section as follows (Page 1, Lines 14-16):
“The purpose of this study was to identify the microbial community of raw milk samples before and after UV-C irradiation and to establish fundamental data on UV-C treatment to improve the safety and shelf life of raw milk.”
2) Comment: A weak discussion on how psychrotrophic bacteria can cause either quality or safety issues to raw milk.
Response: We appreciate the reviewer’s comment. We have added a more detailed discussion on how psychrotrophic bacteria affect raw milk quality and safety. We have addressed this point in the introduction section as follows (Page 2, Lines 43–50).
“These bacteria can be eliminated through pasteurization or ultra-high temperature (UHT) treatment; however, they produce heat-resistant enzymes, such as proteases and lipases, during refrigerated storage [1]. These enzymes are the primary spoilage enzymes affecting the quality of dairy products. Lipases hydrolyze milk triglycerides, leading to lipid oxidation and rancidity, which result in flavor deterioration in products such as cheese and milk. Proteases hydrolyze casein fractions, contributing to milk gelation and bitterness [2,3]. Therefore, controlling the initial load of psychrotrophic bacteria is essential to minimize the negative impact of these spoilage enzymes on dairy product quality.”
- Yuan, L.; Sadiq, F. A.; Burmølle, M.; Wang, N. I.; He, G. Insights into psychrotrophic bacteria in raw milk: A review. Journal of food protection, 2019, 82(7), 1148-1159. https://doi.org/10.4315/0362-028X.JFP-19-032
- Júnior, J. R.; De Oliveira, A. M.; Silva, F. D. G.; Tamanini, R.; De Oliveira, A. L. M.; Beloti, V. The main spoilage-related psychrotrophic bacteria in refrigerated raw milk. Journal of Dairy Science, 2018, 101(1), 75-83. https://doi.org/10.4315/0362-028X.JFP-19-032
- Li, H.; Zhang, Y.; Yuan, X.; Liu, S.; Fan, L.; Zheng, X.; Jiao, X. Microbial biodiversity of raw milk collected from Yangzhou and the heterogeneous biofilm‐forming ability of Pseudomo-nas. International Journal of Dairy Technology, 2023, 76(1), 51-62. https://doi.org/10.1111/1471-0307.12905
3) Comment: For Ultraviolet (UV) radiation, give any references to support its inactivation in pathogenic bacteria and fungi.
Response: We appreciate the reviewer’s comment. We have added references to support the inactivation of pathogenic bacteria in raw milk by ultraviolet (UV) radiation. We have addressed this point in the introduction section as follows (Page 2, Lines 61–70).
“Previous studies have reported that UV-C light can effectively reduce pathogenic bacteria in raw milk and dairy products. Engin B & Karagul Yuceer Y. (2012) demonstrated that UV-C treatment resulted in a significant reduction of approximately 2 log CFU/mL in the total coliform count, Escherichia coli, and Staphylococcus spp. in raw milk, making it as effective as thermal treatment [4]. Similarly, the study by Crook J. et al. (2012) evaluated the effect of UV-C treatment on the inactivation of seven milk-borne pathogens. Among these, Listeria monocytogenes exhibited the highest resistance to UV-C, while Staphylococcus aureus was the most susceptible [5]. To the best of our knowledge, no studies have yet investigated the effects of UV-C treatment on the reduction of Pseudomonas and lactic acid bacteria in milk.”
- Engin, B & Karagul Yuceer, Y. Effects of ultraviolet light and ultrasound on microbial quality and aroma‐active components of milk. Journal of the Science of Food and Agriculture, 2012, 92(6), 1245-1252. https://doi.org/10.1002/jsfa.4689
- Crook, J. A.; Rossitto, P. V.; Parko, J.; Koutchma, T.; Cullor, J. S. Efficacy of ultraviolet (UV-C) light in a thin-film turbulent flow for the reduction of milkborne pathogens. Foodborne path-ogens and disease, 2015, 12(6), 506-513. https://doi.org/10.1089/fpd.2014.1843
4) Comment: Next-generation sequencing (NGS), too much description of its basic information. I would suggest the authors to provide exact applications in the food industry
Response: We appreciate the reviewer’s comment. We have revised the NGS section to focus more on its specific applications in the food industry rather than general background information. We have addressed this point in the introduction section as follows (Page 2, Lines 73–79):
“NGS based metagenomic analysis has been applied in various research fields, including the food industry. In food production, it is used to investigate changes in microbial communities during raw material processing and manufacturing, helping to evaluate microbial diversity and monitor foodborne pathogens [6]. Additionally, in fermented foods, meta-genomic analysis has been utilized to identify key microbial communities during fermentation, enabling the optimization of fermentation conditions and the enhancement of product quality [7].”
- Peruzy, M. F.; Murru, N.; Yu, Z.; Cnockaert, M.; Joossens, M.; Proroga, Y. T. R.; Houf, K. De-termination of the microbiological contamination in minced pork by culture dependent and 16S amplicon sequencing analysis. International journal of food microbiology, 2019, 290, 27-35.
- Piao, H.; Hawley, E.; Kopf, S.; DeScenzo, R.; Sealock, S.; Henick-Kling, T.; Hess, M. Insights into the bacterial community and its temporal succession during the fermentation of wine grapes. Frontiers in microbiology, 2015, 6, 809. https://doi.org/10.1016/j.ijfoodmicro.2018.09.025
5) Comment: In this part, the authors missed some key references to support." Raw milk refrigerated for 14 days, in my opinion, the refrigeration time is too long, 14 d is not widely used in the industry.
Response: We appreciate the reviewer’s comment. We acknowledge that a 14-day refrigeration period may exceed typical industrial practices. However, this duration was intentionally selected to observe more pronounced microbial changes over time and to evaluate the long-term effects of refrigeration in combination with UV-C treatment.
Additionally, the extended storage period allowed us to assess whether UV-C treatment of raw milk could effectively mitigate microbial growth over time, ultimately contributing to the establishment of optimal treatment conditions. Similar approaches have been employed in previous studies examining microbial dynamics in refrigerated milk (Vithanage NR et al., 2017). We believe that these findings offer valuable insights into the potential application of UV-C treatment for enhancing the microbial safety of raw milk during storage.
- Vithanage, N. R.; Dissanayake, M.; Bolge, G.; Palombo, E. A.; Yeager, T. R.; Datta, N. Micro-biological quality of raw milk attributable to prolonged refrigeration conditions. Journal of Dairy Research, 2017, 84(1), 92-101. https://doi.org/10.1017/S0022029916000728
6) Comment: 2.4. isolation and identification of microorganisms in raw milk, the culture conditions are not clear, like incubation time, temperature
Response: We appreciate the reviewer’s comment. We followed the recommended incubation conditions for each selective medium to ensure optimal microbial growth. Specifically, Pseudomonas agar was incubated at 25°C for 48 hours, M17 agar at 37°C for 24 hours, and MRS agar at 37°C for 48 hours. These conditions were chosen based on standard protocols to selectively promote the growth of target microorganisms. We have addressed this point in the materials and methods section as follows (Page 4, Lines 154–155):
“Pseudomonas agar plate was incubated at 25°C for 48 hours, while M17 agar and
MRS agar plates were incubated at 37°C for 24 hours and 48 hours, respectively.”
7) Comment: UV-C Treatment, any reference to support the parameters?
Response: We appreciate the reviewer’s comment regarding the UV-C treatment parameters. The conditions applied in our study were based on previously published research to ensure scientific validity. Specifically, we referred to the study by Kwon et al. (2022), which investigated the inactivation of Bacillus cereus ATCC 14579 in milk samples using UV-C treatment. This study provided a foundation for selecting appropriate UV-C parameters, ensuring their effectiveness in microbial inactivation.
- Kwon, S. W.; Kwon, E. A.; Hong, Y. G.; Kim, S. S. Germination of Bacillus cereus ATCC 14579 spore at various conditions and inactivation of the germinated cells with microwave heat-ing and UVC treatment in milk samples. Lwt, 2022, 154, 112702. https://doi.org/10.1016/j.lwt.2021.112702
8) Comment: Remove any reference in this part to the discussion, simply clarify the data. Discussion part is missed in this manuscript.
Response: We appreciate the reviewer’s comment. We have revised the manuscript accordingly by integrating the missing discussion into the Results section and restructuring it as 'Results and Discussion' to ensure a more comprehensive and coherent presentation of the data.
9) Comment: Carefully check the writing of each bacterial name. Only Latilactobacillus curvatus DSM 20019 was sequenced in this study, need to be clarified.
Response: We appreciate the reviewer’s comments. We have carefully reviewed the manuscript and have added the information in the revised manuscript that the Latilactobacillus curvatus DSM 20019 strain, isolated and identified on MRS agar, which is a selective medium for lactobacilli such as Lactobacillus spp., was used as a lactic acid bacterium in the study. We have addressed this point in the results and discussion section as follows (Page 6, Lines 244–247).
“Therefore, the strain Latilactobacillus curvatus DSM 20019, which was isolated and identified from MRS agar (a selective medium for lactic acid bacteria like Lactobacillus spp.), was used in the study as a lactic acid bacterium.”
10) Comment: Though the authors stated that Post- 172 hoc analysis was performed using Duncan's multiple range test to determine significant differences (p < 0.05), but actually they missed the significant differences analysis in this part.
Response: We appreciate the reviewer’s comment. We conducted post-hoc analysis using Duncan’s multiple range test to determine significant differences among time points for each bacterial strain. The reduction patterns differed among the bacterial strains, with a significant decrease observed initially, followed by inconsistent values over time. This phenomenon may be attributed to the UV-C treatment mechanism, which causes damage rather than complete eradication, potentially affecting the bacterial count. The different superscripts (a, b, c, etc.) in Table 1 indicate statistically significant differences (p < 0.05). To clarify this further, we will explicitly state in the figure legend that different letters represent significant differences between time points within the same bacterial strain.

Reviewer 2 Report
Comments and Suggestions for Authors
The article entitled “Metagenomic analysis of raw milk and the inactivation of food-3 borne pathogen using ultraviolet-C” submitted for consideration by Juhui Lee, Hyeonjun Moon, Ji-In Noh, Hye-Rim Park, Sang-Soon Kim is interesting because the article address important subject represented by food safety and microbiological research. Also, the study investigates the microbial composition of raw milk before and after UV-C irradiation using metagenomic analysis. Researchers collected raw milk samples, stored them at 4°C for 14 days, and applied 5 minutes of UV-C treatment.
Even if it is interesting, in this form I think that there are a few queries that needs to be addressed before processing:
Query 1: After a detailed analyses of the article I identified a few limitations. For example, the study primarily focuses on Pseudomonas spp., Lactococcus spp., Staphylococcus spp., but raw milk contains other significant bacteria, including potential spore-forming bacteria that may be more resistant. Why did you not test for other pathogens?
Query 2: Another major concern that I have is represented by the number of samples. The study analyzed milk from six cows, divided into three groups (two cows per group). However, there is no discussion on whether this sample size is statistically representative of larger dairy populations. I’m wondering how was the sample size determined? Was a statistical analysis performed to ensure representativeness?
Query 3: I know that the limitation and elimination of various pathogens is very important but at the same time, sterilization methods should ensure that the main components of milk are not denatured. Regarded this aspect, I’m wondering did you analyze whether UV-C treatment altered the pH, protein structure, or lipid composition of milk?
Query 4: In my view, any study must have a practical impact for the everyday world. Even if we talk about a research, the practical implications of the results should be identified and discussed in the discussion section. For example, what would be the benefits or challenges that would be encountered when implementing such system at industrial scale? Or how does UV-C compare in cost-effectiveness to traditional pasteurization methods?
Please totally remake the “Reference list”. It is not in accordance with the Journal Guidelines for this section. Please consult the following link: https://www.mdpi.com/journal/foods/instructions
Another aspect is represented by the scientific style that you used. I identified mistakes and way of expression that are not good to be used in scientific style (L:79 “Milking was conducted under identical conditions”). I strongly recommend English editing of the article by a native speaker.
I hope that my queries and suggestion can help the authors to improve their work.
Last but not least I want to wish them good luck in future research projects.
Comments on the Quality of English LanguageEnglish editing should be performed!
Author Response
Responses to the Reviewer’s comments
Dear reviewers and editorial staffs in Foods
We are sincerely grateful for your thorough consideration and scrutiny of our manu-script, “Metagenomic analysis of raw milk and the inactivation of foodborne pathogen using ultraviolet-C”, control number foods-3547901. The manuscript was edited by MDPI author services and certificate was attached.
We have carefully considered all the feedback and made the necessary revisions to the manuscript. Our point-by-point responses to each of the reviewers’ comments are provided below. The reviewers' comments are highlighted in blue, our responses are in black, and the changes to the manuscript are highlighted in red for your convenience. We have made every effort to address all of the reviewers’ concerns and hope that the revised manuscript is now acceptable for publication in Foods.
Reviewer #2
1) Comment: The study primarily focuses on Pseudomonas spp., Lactococcus spp., Staphylococcus spp., but raw milk contains other significant bacteria. Why did you not test for other pathogens?
Response: We appreciate the reviewer’s comment. Analyzing additional pathogenic microorganisms would provide a more comprehensive assessment of the microbiological safety of raw milk. However, our study focused on identifying Pseudomonas spp., Lactococcus spp., and Staphylococcus spp. as the predominant bacterial groups in raw milk, based on metagenomic analysis. We also observed that in UV-C treated refrigerated milk samples (CB), Pseudomonas spp. increased by 29.89%, whereas Lactococcus spp. decreased by 7.72% compared to untreated refrigerated milk (SB). This led us to investigate whether Pseudomonas spp. exhibits greater resistance to UV-C treatment than Lactococcus spp. Therefore, in this study evaluated the resistance of strains during UV-C treatment using Pseudomonas spp., which directly affect the quality of refrigerated raw milk, Staphylococcus spp., known to originate from bovine mastitis, Lactococcus spp., which influence the sensory properties of raw milk, and
Latilactobacillus spp., isolated from actual raw milk samples.
2) Comment: Another major concern that I have is represented by the number of samples. The study analyzed milk from six cows, divided into three groups (two cows per group). However, there is no discussion on whether this sample size is statistically representative of larger dairy populations. I’m wondering how was the sample size determined? Was a statistical analysis performed to ensure representativeness?
Response: Thank you for your thoughtful comment regarding the sample size used in this study. We acknowledge that increasing the sample size would ideally provide a better representation of larger dairy populations. However, due to practical constraints in the laboratory, obtaining a larger quantity of milk from more cows was not feasible. While we understand that a sample size of six cows may not fully represent a large dairy population from a statistical standpoint, we aimed to address this limitation by employing random sampling within the available resources. Specifically, we selected two cows from each group, and for each cow, 200 mL of milk was collected and randomly mixed. This approach was intended to introduce a level of randomness and improve representativeness within the scope of our available resources.
3) Comment: I know that the limitation and elimination of various pathogens is very important but at the same time, sterilization methods should ensure that the main components of milk are not denatured. Regarded this aspect, I’m wondering did you analyze whether UV-C treatment altered the pH, protein structure, or lipid composition of milk?
Response: We appreciate the reviewer’s comment. In this study, our primary focus was to identify the major microbial communities through metagenomic analysis and evaluate the effectiveness of UV-C treatment in reducing pathogens. While we acknowledge the importance of ensuring that the key components of milk remain unaffected, we did not specifically analyze changes in pH, protein structure, or lipid composition in this study. However, we recognize that these factors play a crucial role in maintaining milk quality, and we plan to investigate them in future research to provide a more comprehensive assessment of the impact of UV-C treatment.
4) Comment: In my view, any study must have a practical impact for the everyday world. Even if we talk about a research, the practical implications of the results should be identified and discussed in the discussion section. For example, what would be the benefits or challenges that would be encountered when implementing such system at industrial scale? Or how does UV-C compare in cost-effectiveness to traditional pasteurization methods?
Response: We appreciate the reviewer’s comment. We agree with your suggestion that the study should have a practical impact. To address this, we have revised the discussion section to concisely explore the practical applications of our findings and propose future research directions for implementing this system at an industrial scale. We have addressed this point in the results and discussion section as follows (Page 9, Lines 303–310).
“The major challenges of UV-C treatment is its limited penetration depth, particularly in opaque or turbid liquids such as defatted milk. The high scattering and absorption coefficients of these fluids make it difficult to achieve a uniform fluence distribution, which can affect microbial inactivation efficiency. Consequently, when implementing UV-C treatment on an industrial scale, careful system design is required to ensure effective exposure. This includes incorporating continuous agitation mechanisms and optimized reactor configurations, such as thin-film flow systems or pulsed UV technology.”
5) Comment: Please totally remake the “Reference list”. It is not in accordance with the Journal Guidelines for this section.
Response: We appreciate the reviewer’s comment. We have carefully reviewed the formatting requirements provided in the link and revised the reference list to fully comply with the Foods journal guidelines.
6) Comment: Another aspect is represented by the scientific style that you used. I identified mistakes and way of expression that are not good to be used in scientific style (L:79 “Milking was conducted under identical conditions”). I strongly recommend English editing of the article by a native speaker.
Response: We appreciate the reviewer’s comment. We have carefully reviewed and refined the scientific writing throughout the manuscript. Additionally, we have sought professional English editing by a native speaker to ensure clarity, accuracy, and adherence to scientific writing standards.
Newly added references
- Yuan, L.; Sadiq, F. A.; Burmølle, M.; Wang, N. I.; He, G. Insights into psychrotrophic bacteria in raw milk: A review. Journal of food protection, 2019, 82(7), 1148-1159. https://doi.org/10.4315/0362-028X.JFP-19-032
- Júnior, J. R.; De Oliveira, A. M.; Silva, F. D. G.; Tamanini, R.; De Oliveira, A. L. M.; Beloti, V. The main spoilage-related psychrotrophic bacteria in refrigerated raw milk. Journal of Dairy Science, 2018, 101(1), 75-83. https://doi.org/10.4315/0362-028X.JFP-19-032
- Li, H.; Zhang, Y.; Yuan, X.; Liu, S.; Fan, L.; Zheng, X.; Jiao, X. Microbial biodiversity of raw milk collected from Yangzhou and the heterogeneous biofilm‐forming ability of Pseudomonas. International Journal of Dairy Technology, 2023, 76(1), 51-62. https://doi.org/10.1111/1471-0307.12905
- Engin, B & Karagul Yuceer, Y. Effects of ultraviolet light and ultrasound on microbial quality and aroma‐active components of milk. Journal of the Science of Food and Agriculture, 2012, 92(6), 1245-1252. https://doi.org/10.1002/jsfa.4689
- Crook, J. A.; Rossitto, P. V.; Parko, J.; Koutchma, T.; Cullor, J. S. Efficacy of ultraviolet (UV-C) light in a thin-film turbulent flow for the reduction of milkborne pathogens. Foodborne pathogens and disease, 2015, 12(6), 506-513. https://doi.org/10.1089/fpd.2014.1843
- Peruzy, M. F.; Murru, N.; Yu, Z.; Cnockaert, M.; Joossens, M.; Proroga, Y. T. R.; Houf, K. Determination of the microbiological contamination in minced pork by culture dependent and 16S amplicon sequencing analysis. International journal of food microbiology, 2019, 290, 27-35.
- Piao, H.; Hawley, E.; Kopf, S.; DeScenzo, R.; Sealock, S.; Henick-Kling, T.; Hess, M. Insights into the bacterial community and its temporal succession during the fermentation of wine grapes. Frontiers in microbiology, 2015, 6, 809. https://doi.org/10.1016/j.ijfoodmicro.2018.09.025
- Vithanage, N. R.; Dissanayake, M.; Bolge, G.; Palombo, E. A.; Yeager, T. R.; Datta, N. Microbiological quality of raw milk attributable to prolonged refrigeration conditions. Journal of Dairy Research, 2017, 84(1), 92-101. https://doi.org/10.1017/S0022029916000728
- Kwon, S. W.; Kwon, E. A.; Hong, Y. G.; Kim, S. S. Germination of Bacillus cereus ATCC 14579 spore at various conditions and inactivation of the germinated cells with microwave heating and UVC treatment in milk samples. Lwt, 2022, 154, 112702. https://doi.org/10.1016/j.lwt.2021.112702

Reviewer 3 Report
Comments and Suggestions for Authors
This manuscript describes the reduction by UV-C treatment of bacteria found in raw milk. The methodology is sound, the results make sense, and the conclusions are supported. The paper could be improved by taking the following comments into account.
Abstract: Lines 16-19 are confusing to the reader. Four "dominant genera of raw milk" are listed in lines 16-18, and two "dominant microorganisms in raw milk" not found in those genera are listed in lines 18-19.
2nd paragraph of Introduction: Mention that UV-C has been used on raw milk previously. For instance, a 2021 study by Atik and Gumus (LWT 110322) reduced four pathogenic species in raw milk with UV-C.
Line 206 to the end: The genus and species names should be italicized as in the rest of the paper.
Lines 221-235 are not necessary since it repeats data shown in Table 1. You can point out that that Figure 3 and Table 1 illustrate the decreases with time and that they were different for each bacteria.
Author Response
Responses to the Reviewer’s comments
Dear reviewers and editorial staffs in Foods
We are sincerely grateful for your thorough consideration and scrutiny of our manu-script, “Metagenomic analysis of raw milk and the inactivation of foodborne pathogen using ultraviolet-C”, control number foods-3547901. The manuscript was edited by MDPI author services and certificate was attached.
We have carefully considered all the feedback and made the necessary revisions to the manuscript. Our point-by-point responses to each of the reviewers’ comments are provided below. The reviewers' comments are highlighted in blue, our responses are in black, and the changes to the manuscript are highlighted in red for your convenience. We have made every effort to address all of the reviewers’ concerns and hope that the revised manuscript is now acceptable for publication in Foods.
Reviewer #3
1) Comment: Abstract: Lines 16-19 are confusing to the reader. Four "dominant genera of raw milk" are listed in lines 16-18, and two "dominant microorganisms in raw milk" not found in those genera are listed in lines 18-19.
Response: We appreciate the reviewer’s comment. We have restructured the abstract for improved clarity and logical flow. The term "dominant microorganisms in raw milk" has been revised to "microorganisms in raw milk" to better reflect the goal of isolating microorganisms from the raw milk. We have addressed this point in the abstract section as follows (Page 1, Lines 20).
2) Comment: 2nd paragraph of Introduction: Mention that UV-C has been used on raw milk previously. For instance, a 2021 study by Atik and Gumus (LWT 110322) reduced four pathogenic species in raw milk with UV-C.
Response: We appreciate the reviewer’s valuable comment. In response, we have revised the second paragraph of the Introduction to include previous studies on the application of UV-C treatment in raw milk. We have addressed this point in the introduction section as follows (Page 2, Lines 43–50):
“Previous studies have reported that UV-C light can effectively reduce pathogenic bacteria in raw milk and dairy products. Engin, B., & Karagul Yuceer, Y. (2012) demonstrated that UV-C treatment resulted in a significant reduction of approximately 2 log CFU/mL in the total coliform count, Escherichia coli, and Staphylococcus spp. in raw milk, making it as effective as thermal treatment. Similarly, the study by Crook, J. et al. (2012) evaluated the effect of UV-C treatment on the inactivation of seven milk-borne pathogens. Among these, Listeria monocytogenes exhibited the highest resistance to UV-C, while Staphylococcus aureus was the most susceptible. To the best of our knowledge, no studies have yet investigated the effects of UV-C treatment on the reduction of Pseudomonas and lactic acid bacteria in milk.”
3) Comment: Line 206 to the end: The genus and species names should be italicized as in the rest of the paper.
Response: We appreciate the reviewer’s comment. We have revised the manuscript and italicized the genus and species names from line 206 to the end, as per your suggestion.
4) Comment: Lines 221-235 are not necessary since it repeats data shown in Table 1. You can point out that that Figure 3 and Table 1 illustrate the decreases with time and that they were different for each bacteria.
Response: We appreciate your careful review of our manuscript. We have removed lines 221-235 to avoid redundancy with Table 1. Instead, we have revised the text to concisely indicate that Figure 3 and Table 1 illustrate the decrease over time and the differences observed among the bacterial strains. We have addressed this point in the results and discussion section as follows (Page 7, Lines 255–262):
“Bacterial populations decreased over time due to UV-C treatment, but the extent of reduc-tion varied among the bacterial species. As shown in Table 1, the population of P. aeruginosa decreased by approximately 2 log CFU/mL after 6 minutes of treatment compared to the untreated control, with no significant reduction observed beyond this point. For S. aureus and L. curvatus, a gradual decrease in population was observed over time; however, no significant reduction was noted after 8 minutes of treatment. Meanwhile, L. lactis exhibited a notable reduction, with its count falling below the detection limit (1 log CFU/mL) after 10 minutes of treatment.”
Newly added references
- Yuan, L.; Sadiq, F. A.; Burmølle, M.; Wang, N. I.; He, G. Insights into psychrotrophic bacteria in raw milk: A review. Journal of food protection, 2019, 82(7), 1148-1159. https://doi.org/10.4315/0362-028X.JFP-19-032
- Júnior, J. R.; De Oliveira, A. M.; Silva, F. D. G.; Tamanini, R.; De Oliveira, A. L. M.; Beloti, V. The main spoilage-related psychrotrophic bacteria in refrigerated raw milk. Journal of Dairy Science, 2018, 101(1), 75-83. https://doi.org/10.4315/0362-028X.JFP-19-032
- Li, H.; Zhang, Y.; Yuan, X.; Liu, S.; Fan, L.; Zheng, X.; Jiao, X. Microbial biodiversity of raw milk collected from Yangzhou and the heterogeneous biofilm‐forming ability of Pseudomonas. International Journal of Dairy Technology, 2023, 76(1), 51-62. https://doi.org/10.1111/1471-0307.12905
- Engin, B & Karagul Yuceer, Y. Effects of ultraviolet light and ultrasound on microbial quality and aroma‐active components of milk. Journal of the Science of Food and Agriculture, 2012, 92(6), 1245-1252. https://doi.org/10.1002/jsfa.4689
- Crook, J. A.; Rossitto, P. V.; Parko, J.; Koutchma, T.; Cullor, J. S. Efficacy of ultraviolet (UV-C) light in a thin-film turbulent flow for the reduction of milkborne pathogens. Foodborne pathogens and disease, 2015, 12(6), 506-513. https://doi.org/10.1089/fpd.2014.1843
- Peruzy, M. F.; Murru, N.; Yu, Z.; Cnockaert, M.; Joossens, M.; Proroga, Y. T. R.; Houf, K. Determination of the microbiological contamination in minced pork by culture dependent and 16S amplicon sequencing analysis. International journal of food microbiology, 2019, 290, 27-35.
- Piao, H.; Hawley, E.; Kopf, S.; DeScenzo, R.; Sealock, S.; Henick-Kling, T.; Hess, M. Insights into the bacterial community and its temporal succession during the fermentation of wine grapes. Frontiers in microbiology, 2015, 6, 809. https://doi.org/10.1016/j.ijfoodmicro.2018.09.025
- Vithanage, N. R.; Dissanayake, M.; Bolge, G.; Palombo, E. A.; Yeager, T. R.; Datta, N. Microbiological quality of raw milk attributable to prolonged refrigeration conditions. Journal of Dairy Research, 2017, 84(1), 92-101. https://doi.org/10.1017/S0022029916000728
- Kwon, S. W.; Kwon, E. A.; Hong, Y. G.; Kim, S. S. Germination of Bacillus cereus ATCC 14579 spore at various conditions and inactivation of the germinated cells with microwave heating and UVC treatment in milk samples. Lwt, 2022, 154, 112702. https://doi.org/10.1016/j.lwt.2021.112702

Round 2
Reviewer 1 Report
Comments and Suggestions for Authors
can be accepted.
Reviewer 2 Report
Comments and Suggestions for Authors
Dear authors,
Thank you for your responses.
Please include in the main manuscript your answers especially at Comm. 2 and 3. Those are very important limitations and the answers an pointing them out is mandatory!
Thank you and wish you good luck in the future research projects!
Author Response
Reviewer #2
Dear authors,
Thank you for your responses.
Please include in the main manuscript your answers especially at Comm. 2 and 3. Those are very important limitations and the answers an pointing them out is mandatory!
Thank you and wish you good luck in the future research projects!
We sincerely appreciate for reviewing our revised manuscript. The manuscript was revised once more to include the limitations and further study in the manuscript (L 198-204, L293-297). Please check the revised manuscript once more.
